# DNA-Modified Liquid Crystal Droplets

**DOI:** 10.3390/bios12050275

**Published:** 2022-04-27

**Authors:** Xiuxiu Yang, Xiao Liang, Rajib Nandi, Yi Tian, Yiyang Zhang, Yan Li, Jingsheng Zhou, Yuanchen Dong, Dongsheng Liu, Zhengwei Zhong, Zhongqiang Yang

**Affiliations:** 1Key Laboratory of Organic Optoelectronics and Molecular Engineering of the Ministry of Education, Department of Chemistry, Tsinghua University, Beijing 100084, China; yxx17@mails.tsinghua.edu.cn (X.Y.); liangxiao@tsinghua.edu.cn (X.L.); rajibnandi1987@gmail.com (R.N.); crystal_yi_work@163.com (Y.T.); yiyang.zhang@jiahua-china.com (Y.Z.); li_yan850603@163.com (Y.L.); zhoujingsheng@petrochina.com.cn (J.Z.); dongyc@iccas.ac.cn (Y.D.); liudongsheng@mail.tsinghua.edu.cn (D.L.); 2Department of Chemical Engineering, Hebei Petroleum University of Technology, Chengde 067000, China

**Keywords:** liquid crystal droplets, DNA nanotechnology, DNA aptamer, sensors, microcontact printing

## Abstract

In this work, we have combined the advantages of sequence programmability of DNA nanotechnology and optical birefringence of liquid crystals (LCs). Herein, DNA amphiphiles were adsorbed onto LC droplets. A unique phenomenon of LC droplet aggregation was demonstrated, using DNA-modified LC droplets, through complementary DNA hybridization. Further functionalization of DNA-modified LC droplets with a desired DNA sequence was used to detect a wide range of chemicals and biomolecules, such as Hg^2+^, thrombin, and enzymes, through LC droplet aggregation and vice versa, which can be seen through the naked eye. These DNA-modified LC droplets can be printed onto a desired patterned surface with temperature-induced responsiveness and reversibility. Overall, our work is the first to report DNA-modified LC droplet, which provides a general detection platform based on the development of DNA aptamers. Additionally, this work inspires the exploration of surface information visualization combined with microcontact printing.

## 1. Introduction

The self-assembly of nanoscale particles into microscale aggregates is a powerful method to obtain new materials. It has become a potential technology to spontaneously form complex and higher ordered structures from simple building units to obtain the specificity and reversibility of self-assembly [1]. At the same time, how to better control the interaction between building elements is an open issue in scientific research. These building blocks can be hard colloids (such as gold nanoparticles [2,3], silica particles [4,5], quantum dots [6]), soft colloids (such as polymeric particles [7,8], vesicles [9,10,11,12,13], oil in water (O/W) emulsion droplets [14,15,16,17]), or even cells [18]. DNA oligonucleotides, as a specific connecting element, can be designed to assemble sophisticated and complex, high ordered assemblies in solution [19,20]. An effective method to control specific interactions between building blocks is to decorate the single-stranded DNA on these building blocks. This method not only realizes the formation of a crystal structure [21,22], but also the responsiveness and reversibility to various external stimuli, such as temperature [7,23], electrolytes [24,25], surfactants [26], competitive oligonucleotides [27], and the utilization of the secondary structure of DNA oligonucleotides [28]. It is important note that most of these DNA-modified building blocks can be immobilized on the substrate by DNA complementary pairing, or biotin and streptavidin interaction [29,30,31,32]. The emulsion droplets adsorbed on the solid surface with a desired pattern through DNA complementary principles have not been reported before.

Liquid crystals (LCs) are well-suited for the characterization of the interfacial event as they undergo cooperative long-range ordering in response to external surface stimuli [33,34,35]. Amphiphilic molecules have been found to adsorb at the LC–aqueous interface, where the hydrophilic region points to the aqueous phase and the hydrophobic region of amphiphiles couples with LCs, which induces the internal ordering transitions of LCs [36]. Perturbations in the interactions between LCs and amphiphilic molecules in the presence of an external target in the aqueous phase act as an optical signal for thin film LC, and as a change in internal ordering from radial to bipolar configuration (or vice versa) for LC droplets [36,37,38]. In past studies, conventional amphiphiles, such as surfactants, lipids, and amphiphilic polymers, are mostly used to detect chemicals and biomolecules using LCs [39,40]. However, these conventional amphiphilic molecules lack specific interactions with target molecules. Therefore, the design of programmable interaction between amphiphiles and target molecules is desirable for advancement in constructing an ideal sensor using LCs. In this context, a DNA aptamer is an excellent candidate as a specific recognition unit that can bind to a broad range of targets with high affinity and specificity [41,42,43,44]. A few aptamers used as recognition units in LC-based sensors, with good results, have been reported; in these cases, most of the sensors were fabricated either based on the LC–solid interface [45,46,47] or based on the principle of competitive interactions between non-specific amphiphile–aptamer interactions and specific aptamer–target molecule interactions [35,48,49,50,51,52,53,54]. However, DNA amphiphile-modified LC droplets have not been reported, which may provide an opportunity for a new type of sensing in solution. In addition, further immobilization of DNA-modified LC droplets onto the solid surface can be designed as stimuli-responsive surfaces. Therefore, here, we plan to explore LCs as a new optically active soft building block; LC emulsion droplets that have undergone DNA functionalization may respond to specific interactions in the presence of external analytes, and can be programmed to attach onto specific surface patterns with optical output.

In this work, we developed DNA-modified LC droplets for the first time. Taking the advantage of DNA nanotechnologies, such as sequence programmability and binding fidelity, we have designed a direct assembly of desirable DNA sequences for the DNA-modified LC droplets, and their utilization of various specific recognition, through a unique concept of LC droplet aggregation. In addition, we have demonstrated that these DNA-functionalized LC droplets can be attached through specific interactions to a desirable patterned surface and exhibited thermal responsiveness. The new approaches established in this study, i.e., the aggregation of LC droplets and printing of LC droplets, can be used in several practical applications, and further exploration will provide new information to the field of LC-based sensors and devices.

## 2. Materials and Methods

### 2.1. Materials

LC 4-cyano-4′-n-pentylbiphenyl (5CB) was purchased from Jiangsu Hecheng Display Technology Co., Ltd. (Nanjing, China). The EcoRI and BamHI restriction enzymes were purchased from the TaKaRa Biotech company (Dalian, China). Thrombin was purchased from J&K Scientific (Beijing, China). Poly(dimethyl siloxane) (PDMS) was purchased from Dow Corning (Midland, MI, USA). 12-mercaptododecanoic acid was purchased from Sigma (Burlington, MA, USA). Other chemicals were purchased from Tianjing Fuchen Chemical Reagents Factory (Tianjin, China). All DNA sequences are shown in Appendix A, which were synthesized by Beijing Qingke Biological Technology Co., Ltd. (Beijing, China). Glass microscope slides (Fisher’s Finest Premium grade) were purchased from Fisher Scientific (Pittsburgh, PA, USA). The buffer solution used in this work was phosphate-buffered saline (PBS, pH 7.4, 2 mM KH_2_PO_4_, 8 mM Na_2_HPO_4_, 136 mM NaCl, and 2.6 mM KCl).

### 2.2. Synthesis of DNA Amphiphile DNA-C18

The DNA amphiphiles used in the experiment were synthesized using an ABI 394 DNA synthesizer following a standard phosphoramidite DNA synthesis protocol, and purified by HPLC using water/acetonitrile/TEAA (triethylamine acetate buffer, 100 mM, pH 7.0) as eluent [55]. High-resolution mass spectrometry was recorded using a Waters LCT ESI. Water used in all experiments was Millipore Milli-Q deionized water (18.2 MΩ cm^−1^). The cell crusher (Q125 Sonicator) was purchased from QSonica (Newtown, CT, USA). 

### 2.3. Preparation of DNA-Modified LC Droplets

First, 10 μL LC was added to 50 μL 20 μM of DNA-C18 in PBS (pH 7.4) at 25 °C. Then, the mixture was sonicated for 30 s with 20% power using the cell crusher; this produced a cloudy solution of DNA-modified LC droplets.

### 2.4. Aggregation of DNA-Modified LC Droplets

As for the R1 + R2 linker, 40 μM R1 and 40 μM R2 were mixed in equivalent volume for 5 min to form 20 μM the R1 + R2 linker. Then, 5 μL 20 μM R1 + R2 linker was added to 5 μL of DNA-C18-modified LC droplets, resulting in aggregation of LC droplets after 30 s.

### 2.5. Fluorescence Experiment

The “Y” structure DNA was formed first, including three types of single-stranded DNA (S1, S2, and S3), FAM-DNA, and ROX-DNA. The concentration of each S1, S2, and S3 was 60 μM. Subsequently, 10 μL S1, S2, and S3 were mixed together for 15 min. Then, 20 μM 4 μL ROX-DNA and 20 μM 16 μL FAM-DNA was added, and left to stand for 15 min to form the “Y” structure. Once formed, 20 μL “Y” structure was added to 20 μL LC droplets for 15 min~30 min to detect fluorescence signals. The fluorescence experiment was conducted using a Zeiss Positive laser confocal microscope LSM780. The excitation and emission wavelengths of FAM are 492 and 518 nm, and those of ROX are 578 and 604 nm, respectively.

### 2.6. Hg^2+^ Responsiveness of DNA-Modified LC Droplets

First, 10 μL DNA-modified LC droplets and 10 μL 20 μM Hg^2+^ aptamer 1 and 2 were mixed together, to deposit the DNA-C18 on the LC droplets. Then, 20 μL Hg^2+^ was added, the resulting final concentration of Hg^2+^ was 0, 5, 10, 25, and 50 mM.

### 2.7. Thrombin Responsiveness of DNA-Modified LC Droplets

As in the previous subsection, 10 μL DNA-modified LC droplets and 10 μL 20 μM thrombin aptamer 1 and 2 were mixed together, to deposit the DNA-C18 on the LC droplets. Then, 20 μL thrombin was added, the resulting final concentration of thrombin was 0, 1, 5, and 10 mM.

### 2.8. Enzyme Responsiveness of DNA-Modified LC Droplets

We chose three types DNA sequences as linkers: random1/random2, R1/R2, and H1/H2. Random1/random2 included no cutting site, R1/R2 included a cutting site for EcoRⅠ, and H1/H2 included a cutting site for BamHⅠ. For each linker, 10 μL 20 μM linker was added to 10 μL DNA-modified LC droplets, left to stand for 30 min, then EcoRⅠ or BamHⅠ was added. The final concentration of enzyme was 45 U.

### 2.9. Preparation of Gold Substrate

First, the glass slides were cleaned with piranha solution [56]. Then, the piranha-cleaned glass slides were evaporated with 10 nm chromium and 20 nm gold (Tsinghua Foxconn Nanotechnology Research Center, Tsinghua, Beijing, China).

### 2.10. Self-Assembly of 12-Mercaptododecanoic Acid Monolayer

The stamp used in the microcontact printing method was fabricated by PDMS [57]. Stamps were incubated in 20 μL 12-mercaptododecanoic acid alcohol solution for 20 s, and then blown dry with nitrogen. The stamp was printed onto the gold substrate surface and left in place for 30 s.

### 2.11. Formation of Mixed SH-DNA/12-Mercaptododecanoic Acid Monolayer

The printed area was incubated in 1 μM SH-DNA (PBS pH 7.4) for 16 h, then immersed in PBS three times to remove unabsorbed SH-DNA. Complementary DNA (C-DNA-SH) and noncomplementary DNA (NC-DNA-SH) can be modified on the unprinted gold surface. The surface was covered by PBS at all times.

### 2.12. DNA Hybridization on Substrate Surface

The LC droplets were diluted 10 times using PBS solution. The diluted droplets were added onto the SH-DNA/12-mercaptododecanoic acid monolayer for 5 min. PBS was used to wash the unhybridized droplets.

### 2.13. Thermal Responsiveness and Repeatability of LC Patterns

An Au substrate immobilized by LC droplets was immersed in 40 °C PBS solution. As temperature gradually reduced in the PBS solution, changes in the LC droplets could be observed under polarized light. An adsorbed Au substrate, with liquid crystal droplets, was immersed in 60 °C PBS for 1 min, while slightly washing the surface. Adsorption and washing were repeated three times, monitored under a polarizing optical microscope.

### 2.14. Characterization of LC Optical Signals

The optical images were captured using a Nikon Eclipse Ti microscope with Nikon DS-U3 in transmission mode, under a resolution of 1600 × 1200 pixels, a gain of 1.00×, and a shutter speed of 1/30 s.

## 3. Results and Discussion

### 3.1. Preparation of DNA-Modified LC Droplets

In this work, we have chosen a DNA amphiphilic molecule DNA-C18 (as shown in Figure 1) composed of an octadecane alkyl chain and a DNA sequence, employing hydrophobic and hydrophilic regions, respectively. The DNA amphiphilic molecule DNA-C18 was synthesized and purified according to literature procedures [55,58], the mass spectrum is shown in Appendix A. The peak at 5633 corresponds to the molecular weight of DNA, and 5965 corresponds to the total molecular weight of DNA-C18. There are also some DNA impurities and DNA-C18 impurities present. These suggest that the amphiphilic molecule DNA-C18 was successfully synthesized. For the preparation of DNA-modified LC droplets, we used a modified tip sonication method. The emulsions were formed by tip sonication of 10 μL nematic LC 5CB (the structure is shown in Figure 1) in 50 μL of 20 μM DNA-C18 aqueous solution at 25 °C. The obtained emulsion, as shown in Figure 2a–c, can remain stable for several weeks. It is important to mention that the amount of 5CB in the aqueous solution used to prepare the LC emulsion is relatively large compared to the conventional method in the reported literature [59]. This is because DNA-C18 itself tends to form very stable assemblages, such as micelles, which leads to its reluctance to adsorb at the 5CB surface [55,58]. To overcome this difficulty, exploit high-power tip sonication instead of conventionally used relatively low-power bath sonication, and vortex to disassemble DNA-C18 amphiphilic assemblages. The curved structure of the LC droplets also facilitated the adsorption of DNA-C18 amphiphiles. As the high power produced much smaller LC droplets, which were difficult to observe under the optical microscope, a much higher quantity of LCs was used, which generated much larger DNA-modified LC droplets, suitable for further studies (Figure 2a,b).

To prove that the LC droplets were modified with DNA amphiphilic molecules, we utilized a DNA linker to connect with the LC droplets. One type of DNA linker is a combination of two single strands of DNA, R1 and R2, of which both sticky ends of the R1 + R2 linker can hybridize with the DNA sequence on the LC droplets. The mechanism is shown as a diagram in Figure 2. The experimental results demonstrated that DNA-modified LC droplets dispersed homogeneously before adding the R1 + R2 linker (Figure 2a,b); under a polarized microscope, the LC droplets appeared bright due to the optical property of the LCs. The hydrophilicity and negative charges of DNA prevent the collapse of the LC droplets, and thus the obtained DNA-modified LC droplets can remain stable for several days. However, after adding the R1 + R2 linker, the DNA-modified LC droplets aggregated and formed sediments, which can be observed under microscope and with the naked eye, as shown in Figure 2d–f. The aggregation of LC droplets can be attributed to the hybridization between the R1 + R2 linker and DNA deposited on the LC droplets. In order to preclude the effect of the DNA linker on the aggregation of the DNA-modified LC droplets, we also performed a control experiment using either R1 or R2. When either R1 or R2 was incubated with the DNA-modified LC droplets under identical experimental conditions, we observed no changes in the DNA-modified LC droplets, as shown in Appendix A. This result suggests that R1 or R2 alone could not induce any aggregation in the DNA-modified LC droplets, this is because R1 and R2 in isolation did not act as a bridge to connect LC droplets, while the pre-controlled R1 + R2 linker can induce aggregation in DNA-modified LC droplets (Figure 2). The above results not only prove that DNA was deposited on the LC droplets, but also develop a new and unique concept of LC droplet aggregation for specific interactions in LC research.

It is important to mention that the droplet configuration of the DNA-modified LC droplets was seen as a bipolar configuration before as well as after aggregation. We expected that the adsorption behavior of DNA-C18 would resemble that of well-studied amphiphiles, such as surfactants, lipids, and amphiphilic polymers; i.e., the hydrophobic part of the amphiphiles would insert into the LC phase, while the amphiphilic part would stay in the aqueous phase, which would result in the orientational change of the LC droplet from bipolar to radial [36,37]. To further investigate this, we calculated the interfacial number density of DNA-C18 adsorbed onto the surface of the LC droplets by assuming that all DNA-C18 amphiphiles are adsorbed onto the surface of 10 µm LC droplets. The interfacial number density for DNA-C18 on the surface of the LC droplets is ~50 nm^2^/molecule, whereas for single- and double-tailed amphiphiles, the interfacial density for homeotropic ordering is reported to be in the range of 0.4–0.9 nm^2^/molecule [60,61,62]. Therefore, low interfacial density may be the reason for our observation of bipolar configuration of DNA-modified LC droplets. DNA has many negative charges that repel each other; this could also be a factor contributing to the low interfacial density of DNA-C18 on the surface of the LC droplets.

For more direct confirmation of DNA functionalization on the LC droplets, we used confocal microscopy with fluorescence probes. First, the “Y” shape structure of DNA was designed using the fluorescence probes, (as shown in Figure 3), containing three types of single-stranded DNA (S1, S2, and S3), FAM-DNA, and ROX-DNA. Subsequently, DNA deposited on the LC droplets hybridized with the sticky ends of the “Y” shape structure. The two fluorophores, FAM (green, Ex: 492 nm, Em: 518 nm) and ROX (red, Ex: 578 nm, Em: 604 nm) can be visualized via confocal microscope. Figure 3 clearly demonstrates that both FAM and ROX were successfully attached to the DNA-modified droplets through DNA hybridization. This experiment proved that LC droplets were indeed modified with DNA at the surface. In addition, it also provides an indication that we can further decorate the LC droplets with various functional groups.

The demonstration of aggregation of the DNA-modified LC droplets on a pre-programmed DNA linker permitted us to design and develop a new principle for the detection of target chemicals and biomolecules by selecting an appropriate DNA aptamer, which can be a sensitive and easy way to see the optical output through the naked eye. As proof of this concept, we investigated the sensing capability of DNA-C18 modified LC droplets towards the detection of various target analytes, such as mercury ions (Hg^2+^), thrombin, and enzymes, covering most of the existing sensing mechanisms available for aptamer-based sensors [42,44].

### 3.2. The Responsiveness of LC Droplets

#### 3.2.1. Hg^2+^ Responsiveness

Our first target for detection was hazardous Hg^2+^, utilizing DNA-modified LC droplets. Mercury ion is considered one of the most hazardous chemicals to the human body due to its several toxic effects [63]. Aptamers are well-known, regarding the detection of mercury ions, for their selectivity and specificity, and were found to be hybridized in the presence of Hg^2+^ ions. For the detection of Hg^2+^, we designed two aptamers, Hg^2+^ aptamer 1 and aptamer 2. Both of them simultaneously hybridized with the DNA-C18 on the LC droplet. In the presence of Hg^2+^, their remaining exposed sequences can connect to each other through the formation of a T–Hg^2+^–T structure [64]. The mixture of Hg^2+^ aptamer 1 and aptamer 2 was added to the DNA-modified LC droplets. As the aptamers can hybridize with the DNA on the DNA-modified LC droplets, a homogenous dispersion of DNA aptamer-modified LC droplets was obtained, as shown in Figure 4b. Upon addition of 10 mM concentration of Hg^2+^ to the DNA-modified LC droplets, the LC droplets were aggregated, as shown in Figure 4, suggesting that hybridization took place between Hg^2+^ aptamer 1 and aptamer 2 coated on the LC droplets. More LC droplets aggregated together with the increasing concertation of Hg^2+^, and aggregation between the LC droplets was reduced with the decreased concentration of Hg^2+^. These results indicate that DNA-modified LC droplets could detect Hg^2+^ via the aggregation of the LC droplets.

#### 3.2.2. Thrombin Responsiveness

After successful demonstration of Hg^2+^ detection using DNA-modified LC droplets, we decided to use the detection of thrombin as a model for macro-biomolecules, to determine how the DNA-modified LC droplets respond to such a macromolecule. The mechanism of detection of thrombin is as shown in Figure 5, and is similar to that of Hg^2+^ detection using DNA-modified LC droplets. Without adding thrombin, DNA-modified LC droplets were dispersed homogeneously (Figure 5a,b). After adding a small amount of thrombin (1 μM), LC droplets were aggregated together as thrombin recognition aptamers 1 and 2 hybridized with each other in the presence of thrombin (Figure 5c,d) [65,66]. However, at higher concentrations of thrombin, as shown in Figure 5e–h, the aggregation of LC droplets turned out to be inconspicuous. We also noted that the aggregation of LC droplets induced by thrombin is less obvious compared with that of Hg^2+^. The possible reason for this may be because thrombin is a biological macromolecule, and thus its movement rate is expected to be much slower. The slow movement rate may prevent the linking together of the LC droplets, especially at higher concentrations of thrombin, where most of the DNA-modified LC droplets were combined with thrombin. However, this system can be used to detect thrombin at lower concentrations.

#### 3.2.3. Enzyme Responsiveness

To prove that the DNA-modified LC droplet also has enzyme responsiveness, we introduced an enzyme-specific cutting site into the DNA linker sequence [67]. Three different DNA sequence linkers were chosen in the experiment that can hybridize with DNA-C18 and induce the aggregation of DNA-modified LC droplets. These are random1/random2, R1/R2, and H1/H2, which were characterized by no cutting site for enzyme EcoRⅠ and BamHⅠ, a cutting site for enzyme EcoRⅠ, and a cutting site for enzyme BamHⅠ, respectively [68,69]. Before adding the DNA linkers, the DNA-modified LC droplets dispersed homogeneously (Figure 6a). After adding the DNA linkers, random1/random2, R1/R2, and H1/H2, the LC droplets were aggregated together due to the complementarity between the DNA. Once the enzyme was mixed in with the aggregated DNA-modified LC droplets with the corresponding DNA cutting site, the aggregated DNA-modified LC droplets would be cut into single droplets and form a homogeneous emulsion, as shown in Figure 6c,g. However, if the DNA-modified LC droplets were incubated with a non-specific enzyme, they would remain in an aggregated state (Figure 6b,d–f). The change between dispersed and aggregated states indicated that the DNA-modified LC droplet has enzyme responsiveness, which could be further designed to detect enzyme and enzyme activity, both specifically and selectively.

In these systematic studies, we demonstrated that DNA-modified LC droplets could be utilized to detect various types of target molecules upon suitable functionalization of specific DNA through a new type of detection principle; from singlet LC droplets to aggregated droplets, and vice versa, which offers a defect-free approach for LC droplets. The characterization of defect configuration (bipolar or radial) of LC droplets is difficult for micron-sized LC droplets, and needs skilled persons and advanced configured microscopes. However, with the approach used in this work, the sensing output signal (from singlet LC droplets to aggregated droplets, and vice versa) can be visualized with the naked eye without characterizing the configuration of the LC droplets. This approach will also open new opportunities to detect a range of biomolecules using LCs that are unable to induce the transition from bipolar to radial, or vice versa, at the LC–aqueous interface due to their small size or other limitations.

### 3.3. Printing Patterns with DNA-Modified LC Droplets

The demonstration of DNA-modified LC droplets established a new strategy to detect chemicals and biomolecules based on the aggregation of the LC droplets and their reversal phenomena, which motivated us to design a patterned surface modified by DNA-modified LC droplets. This was designed based on an approach such that the introduction of DNA-modified LC droplets onto a substrate surface can bind to the predefined patterned surface through the base pair complementation rule, forming a particular pattern that can be seen under polarized light (Figure 7a–c). We utilized microcontact printing to modify an Au substrate with 12-mercaptododecanoic acid, and then used thiolate DNA to modify the patterned area of the Au substrate. The modification of 12-mercaptododecanoic acid on the area outside of the pattern was to prevent the non-specific adsorption of LC droplets at the Au substrate. Appendix A shows that, when the Au substrate was modified by complementary DNA-SH (C-DNA-SH), the polarized optical signal of LC from a patterned surface was observed under crossed polarized light, which was due to the attachment of DNA-modified LC droplets to the patterned substrate. However, if the Au substrate was modified by non-complementary DNA-SH (NC-DNA-SH), no polarized optical signal of LC droplets was observed, suggesting that LC droplets cannot bind to the substrate. After washing, no LC droplets stayed on the Au surface, which resulted in no optical signals or pattern. These results demonstrate that the DNA-modified LC droplets can be printed on the plane substrate through specific interactions. It is to be noted that after contact was made between the DNA-C18 modified LC droplets and the complementary or non-complementary DNA-functionalized surface, we observed no changes in bipolar droplet configurations; in contrast, Kinsinger et al. reported more aggressive droplet configuration changes in amphiphilic-polymer-coated LC droplets from bipolar/pre-radial to radial configuration, after attaching to the surface through covalent bonding [70].

Next, we carried out a temperature response study of the DNA-modified LC droplets attached to a patterned surface. The DNA-modified LC droplets on the patterned surface were heated to the isotropic phase of LC (40 °C), resulting in the disappearance of the pattern, as shown in Appendix A. Upon cooling from the isotropic phase, the optical birefringence of pattern surface emerged gradually with the decrease in temperature, as these LC droplets were transitioning from the isotropic to nematic phase. This clearly confirms that the optical behavior of the DNA-modified LC droplets on the surface were highly temperature responsive due to the birefringence and thermo-responsive nature of LC.

Furthermore, we performed a repeatability experiment, using the patterned Au substrate, for the adsorption of DNA-modified LC droplets, to check the reusability of the Au pattern surface for effective attachment with LC droplets; this is essential for making a low-cost substrate for practical uses. In this experiment, the DNA-modified LC droplet-functionalized Au substrate was immersed in PBS solution at 60 °C and gently washed with the substrate. Consequently, LC droplets are separated from the substrate, as shown in Figure 7b,e and Appendix A, due to the breaking of the base pair interaction of DNA at 60 °C. Then, the Au substrate was again immersed in the DNA-modified LC droplet solution to attach the DNA-modified LC droplets onto the Au substrate through specific interaction, and consequently, birefringence of the specific pattern remerged under polarized light. This process had been repeated several times, which suggests that the modified Au substrate can be utilized to adsorb DNA-modified LC droplets at least three times without any pattern destruction. The attachment of LC droplets to a specifically patterned Au substrate has tremendous potential for reusability. This method provides new ideas and applications for surface information visualization, the construction of the read/write function, product anti-counterfeiting, and detection kit.

## 4. Conclusions

In this work, we prepared DNA-modified LC droplets using the DNA-C18 amphiphile, which was further functionalized with DNA through specific interactions. Key points of the studies are manyfold; first, this is the first report of the preparation of DNA-modified LC droplets. Second, we can tailor the DNA sequence (such as aptamers) on the surface of DNA-modified droplet through DNA hybridization, which gives us freedom to make a more programmable device. Third, a new and unique concept of LC droplet aggregation in the LC field research was developed, which can be used as a functional response system in the presence of a specific target through specific interactions. Our results suggest that sensors based on DNA-modified LC droplets are fluorescence-label-free, and their optical output can be seen through the naked eye. We also show that the aggregation of LC droplets is a reversible phenomenon (singlet-to-aggregation form, and also from aggregation-to-singlet form) that can be controlled on pre-programmed external target-based specific interactions, and this principle of detection can be applied to all other chemicals and biomolecules that have recognized aptamers. The scope of future work from the LC droplet aggregation phenomenon reported here may create new space for the application of other LC phases (such as cholesteric, blue phase) that might show a more enhanced property on aggregation due to changes in pitch length, and/or reflected visible color [71,72]. Fourth, we demonstrate an effective method for using DNA-modified LC droplets to print a specific pattern onto an Au substrate through specific interactions, which are optically active in their responsiveness to temperature, and are sufficiently reusable. These DNA-functionalized LC droplets assembled on the substrate will be a very efficient and suitable platform for the practical application of LC droplets in surface information visualization and detection kits.

## Figures and Tables

**Figure 1 biosensors-12-00275-f001:**
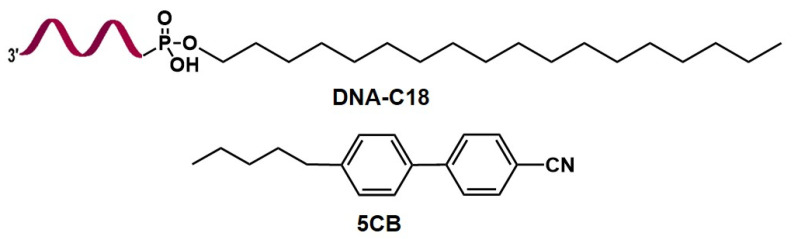
Structures of DNA-C18 and 5CB.

**Figure 2 biosensors-12-00275-f002:**
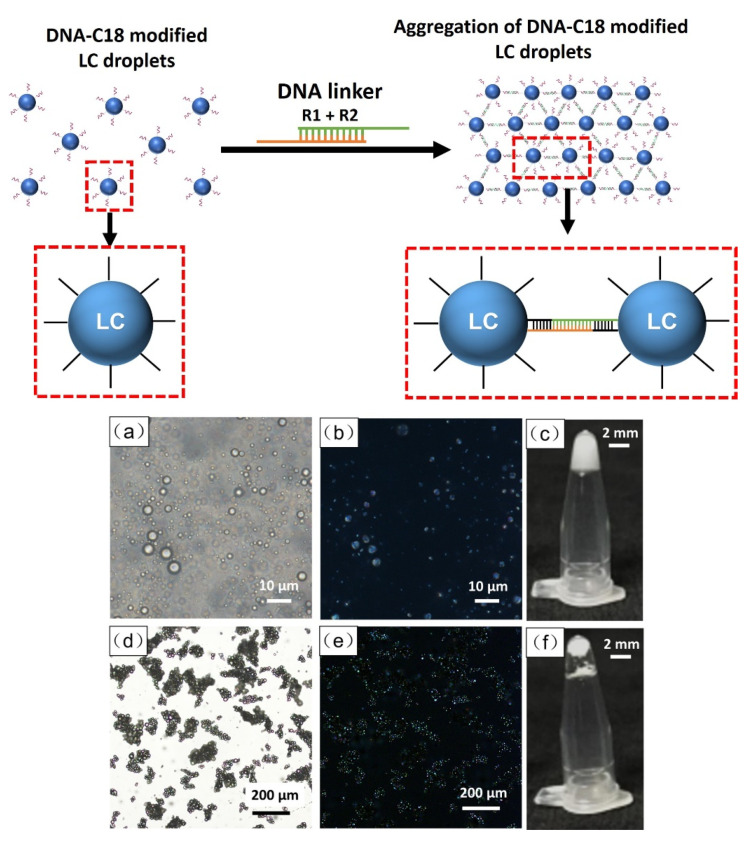
Schematic diagram of DNA-modified LC droplets and aggregated DNA-modified LC droplets in presence of DNA linker R1 + R2. Microscope images of DNA-modified LC droplets before (**a**–**c**) and after (**d**–**f**) adding DNA linker; (**a**,**d**) bright field, (**b**,**e**) polarized light, (**c**,**f**) photograph of macroscopic emulsion.

**Figure 3 biosensors-12-00275-f003:**
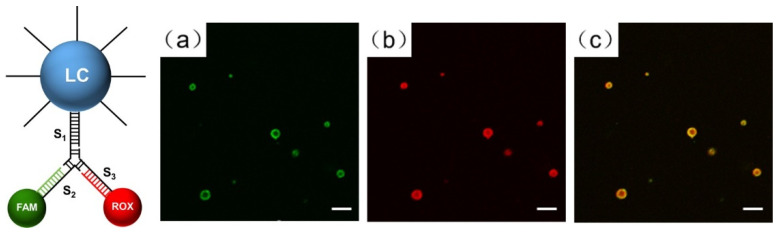
Schematic representation and the confocal images of LC droplets modified by DNA-C18 and hybridized with FAM- and ROX-labeled DNA strands: (**a**) green channel, (**b**) red channel, and (**c**) merged green/red channels. Scale bar is 10 μm.

**Figure 4 biosensors-12-00275-f004:**
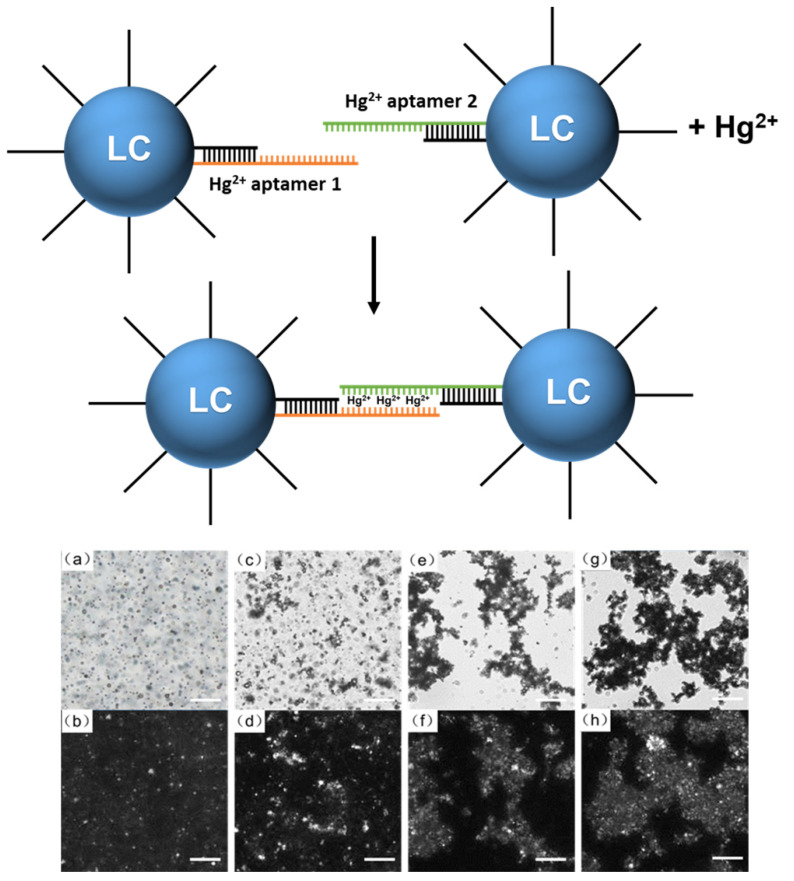
Schematic representation of the responsiveness of DNA-modified LC droplets to Hg^2+^. Optical images of (**a**,**b**) the DNA-modified LC droplets hybridized with Hg^2+^ aptamer 1 and aptamer 2, respectively, and in the presence of (**c**,**d**) 5 mM Hg^2+^, (**e**,**f**) 10 mM Hg^2+^, (**g**,**h**) 15 mM Hg^2+^ in bright field (**a**,**c**,**e**,**g**) and polarized field (**b**,**d**,**f**,**h**). Scale bar is 100 μm.

**Figure 5 biosensors-12-00275-f005:**
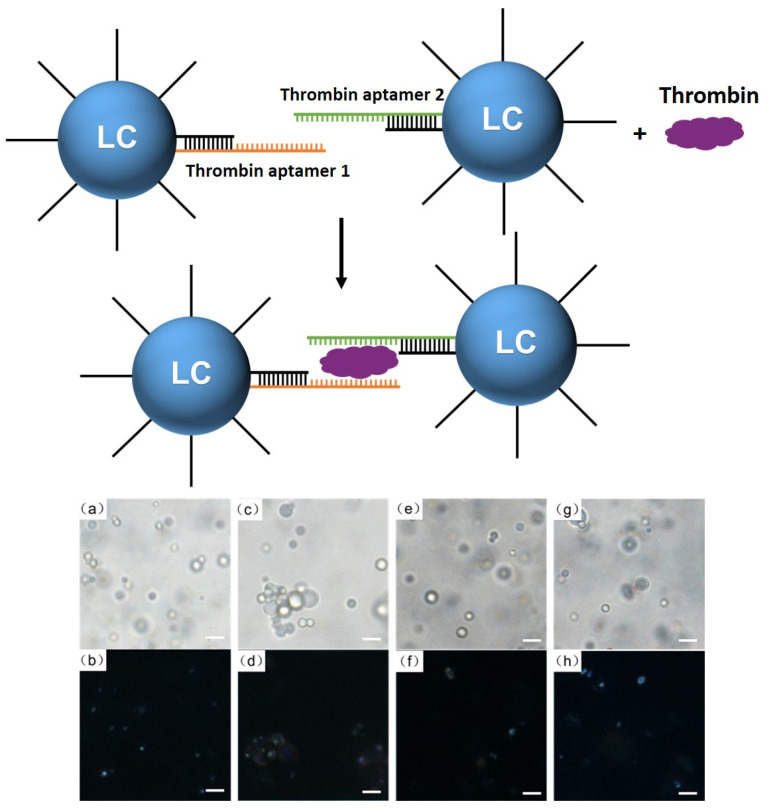
Schematic representation of the responsiveness of LC droplets to thrombin. Optical images of (**a**,**b**) the DNA-modified LC droplets hybridized with thrombin aptamer 1 and aptamer 2, respectively, and in the presence of (**c**,**d**) 1 μM, (**e**,**f**) 5 μM, (**g**,**h**) 10 μM thrombin in bright field (**a**,**c**,**e**,**g**) and polarized field (**b**,**d**,**f**,**h**). Scale bar is 10 μm.

**Figure 6 biosensors-12-00275-f006:**
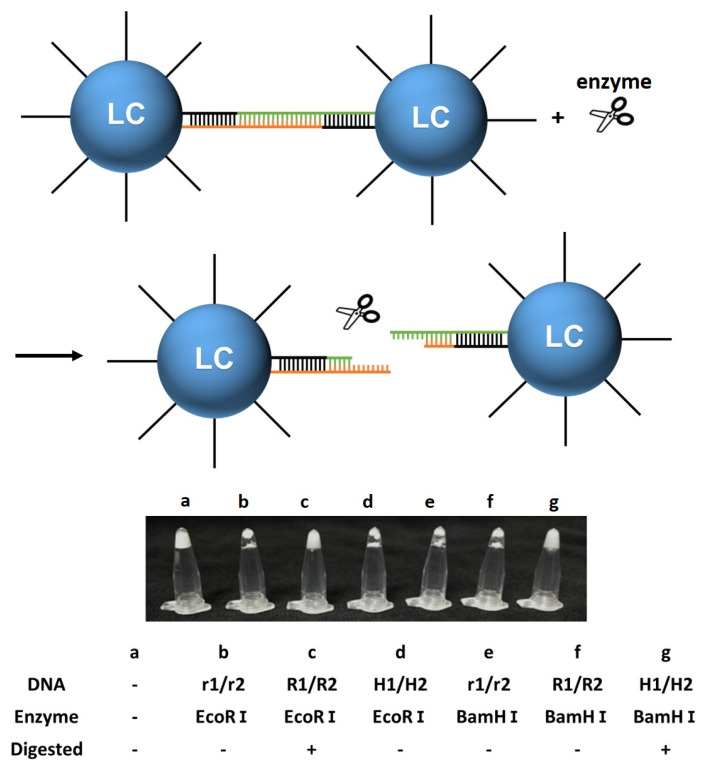
Schematic representation of the responsiveness of LC droplets to enzymes. The photographs of the LC droplets in DNA-C18 with and without enzyme. (**a**) Unmodified LC droplets, (**b**) random1 + random2 + EcoRⅠ, (**c**) R1 + R2 + EcoRⅠ, (**d**) H1 + H2 + EcoRⅠ, (**e**) random1 + random2 + BamHⅠ, (**f**) R1 + R2 + BamHⅠ, (**g**) H1 + H2 + BamHⅠ.

**Figure 7 biosensors-12-00275-f007:**
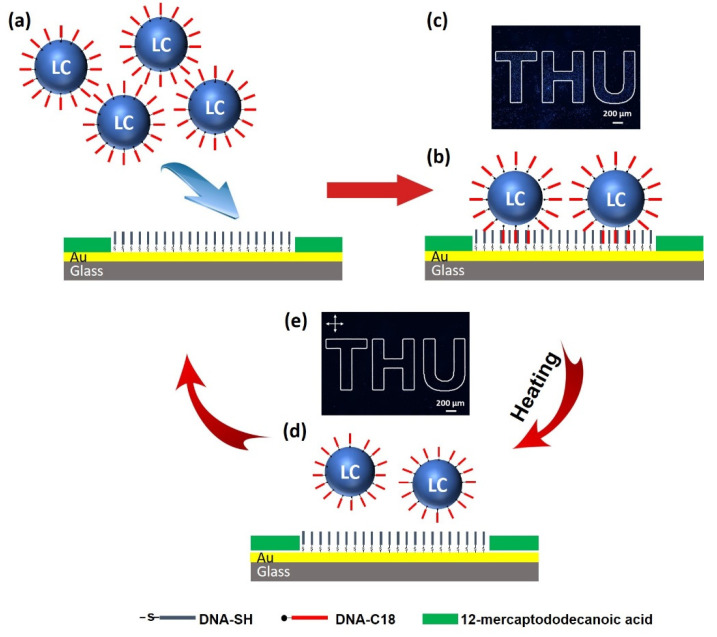
Schematic illustration of repeatable printing patterns with DNA-modified LC droplets. (**a**) Add DNA-modified LC droplets onto the surface of DNA-modified gold substrate, (**b**) the formation of a specific pattern through the base pair complementation rule, (**c**) corresponding polarized image, (**d**) experimental repeatability through breaking the base pair bond at 60 °C, and (**e**) corresponding polarized image taken at room temperature.

## Data Availability

The data presented in this study are available on request from the corresponding author.

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
