# Peer review of "DNA-Modified Liquid Crystal Droplets"

_biosensors, 2022, doi:10.3390/bios12050275_

Round 1

Reviewer 1 Report

In this work, the author demonstrated LC droplets aggregation using DNA modified LC droplets through complementary DNA hybridization, and applied this phenomenon for detection of a wide range of chemicals and biomolecules. Overall, the idea is interesting, and the results are important. I recommend its publication after minor revision of following points.

  • How does the DNA-C18 anchored to the LC surface?
  • The author synthesized DNA-C18 as literature and provided the mass spectrum in Figure S1. It is better to analyze the mass spectrum.
  • In Figure 5, author exhibited the optical images of DNA modified LC droplets for thrombin detection. As author described “After adding a small amount of thrombin (1 μM), LC droplets were aggregated together obviously as thrombin recognition aptamers 1 and 2 were hybridized with each other in presence of thrombin (Figure 5c and d)”. However, the aggregation of LC droplets is not clear enough here.
  • In Figure 6b, d, e, f, why are the DNA modified LC droplets divided into two layers?

Reviewer 2 Report

The authors of "DNA Modified Liquid Crystal Droplets" developed DNA modified LC droplets using DNA amphiphilic complex incorporated onto microscopic droplets of nematic liquid crystal. Utilizing the ability of a single-stranded DNA to hybridize with a complementary strand through base-pairing allows for the specific recognition as observed by the aggregation of LC droplets once the specific sequence of DNA linkers is introduced into the emulsion. This could be used as a detection method for various chemicals and biomolecules and proof of this concept was demonstrated in the manuscript through the detection of mercury and thrombin in the presence of specific DNA recognition aptamers. The presented experimental results, methodology, and protocols are clear and correspond well to the tested hypotheses. However, the treatment of the observation and explanation of the mechanisms lack the ground in the case of LC droplet’s modification of the transmitted polarized light intensity in aggregation and freely suspended state. One could also argue that the detection limit for the presented in this work agents may be extremely high and the use of this method may not be feasible without further improvements. Overall, this work can be published after the authors fully address the comments below.

The observed type of DNA modified LC droplets, as claimed by the authors, is bipolar and so the schematics of the DNA decorating surface of the LC droplets is misleading since that arrangement is clearly radial. Can authors explain the choice of represented structures in Figure 2 and others?

Text on page 11, lines 343, 346, the authors seem to be referring to “birefringence” as the intensity of light passing through the emulsion of DNA-LC droplets placed between two crossed polarizers of the microscope. Those are not the correct statements, since birefringence is the general property of optically anisotropic materials, such as nematic liquid crystals, and what is observed is the intensity of the transmitted light which changes with the reconfiguration or reorientation of the director within droplets (or bipolar axis relative orientation with respect to the polarizers) upon aggregation or surface binding. 

In the same paragraph, it is not clear how binding/unbinding of the DNA-LC droplets to the patterned surface would change the intensity of transmitted light since the authors observed no changes in bipolar droplet configuration or arrangement. What is the reason for the extinction of the pattern once the surface is modified by non-complimentary DNA? The results of extinction of the droplets upon heating are understood, the droplets undergo a transition to optically isotropic non-birefringent phase.

Please also add in the caption for Figure 7 the details of the panels (a)-(e) as well as to the main text. Is “Heating” in Figure 7 referring to the melting of double-stranded DNA and breaking of base-pair interaction at 60°C?

Could the authors explain more about the feasibility and estimate the limit of detection for the described in the manuscript method?

Reviewer 3 Report

Xiuxiu Yang prepared a manuscript on the possibility of using DNA modified LC droplets as biosensors and visualization device. The work is scientifically well developed, even if a check of the englsh has to be done. I suggest the publication in biosensors, after that the authors discuss what I have indicated in the attached files.

Round 2

Reviewer 2 Report

I'm satisfied by the author's responses and changes made to the manuscript